# Morphological, Phaneroptic, Habitat and Population Description of Three Muntjac Species in a Tibetan Nature Reserve

**DOI:** 10.3390/ani12212909

**Published:** 2022-10-24

**Authors:** Yuan Wang, Dajiang Li, Guanglong Wang, Pu Bu Dun Zhu, Wulin Liu, Cheng Li, Kun Jin

**Affiliations:** 1Ecology and Nature Conservation Institute, Chinese Academy of Forestry, Beijing 100091, China; 2Research Institute of Natural Protected Area, Chinese Academy of Forestry, Beijing 100091, China; 3Key Laboratory of Biodiversity Conservation of National Forestry and Grassland Administration, Beijing 100091, China; 4Forestry Inventory and Planning Institute of Tibet Autonomous Region, Lhasa 850000, China; 5Xizijiang Conservation Center, Shenzhen 518000, China

**Keywords:** red muntjac (*Muntiacus vaginalis*), Gongshan muntjac (*Muntiacus gongshanensis*), Fea’s muntjac (*Muntiacus feae*), camera trap, Yarlung Zangbo Grand Canyon National Nature Reserve, habitat, population, conservation, wild species

## Abstract

**Simple Summary:**

Due to the special geographical environment and underdeveloped infrastructure in southern Tibet, China, there are still many wild animals in this area that are not fully understood, especially wild animals with narrow distributions that are rare. Muntjacs are one such type of wild animal, as they have been weakly studied in terms of their physical characteristics, population, and habitat. Based on ten years of field monitoring efforts, this study obtained ample photographic evidence and original data on three species of muntjacs (red muntjac, Gongshan muntjac, and Fea’s muntjac) in a Tibetan nature reserve. Using the obtained data and various models, the characteristics of the muntjac species were distinguished and their relatively objective population numbers and habitat occupancy were determined. This paper answers basic questions about the three different species of muntjac from the aspects of morphology, population sizes and density, and habitat occupation. This information will be useful for researchers and conservation management departments, and it will assist in providing well-informed suggestions for the management of protected areas in this region.

**Abstract:**

Researchers have proposed a variety of classification schemes for the species in the genus *Muntiacus* (Artiodactyla: Cervidae) based on morphological, molecular, and other evidence, but disputes remain. The Tibetan Yarlung Zangbo Grand Canyon National Nature Reserve in the Eastern Himalayas is an area with a rich diversity of muntjac species. The habitats of many species overlap in this area, but systematic research in this area is lacking. To clarify the species, population and habitat size of muntjac species in the study area, we used camera-traps to monitor muntjacs in the nature reserve from 2013 to 2021 and described and compared morphological characteristics of the muntjac species. Subsequently, we used the MaxEnt model to simulate the habitats of the muntjac species and the Random Encounter Model to estimate the population density and numbers of muntjacs. Three muntjac species were found in the area, namely *Muntiacus vaginalis* (n = 7788 ± 3866), *Muntiacus gongshanensis* (n = 6673 ± 2121), and *Muntiacus feae* (n = 3142 ± 942). The red muntjac has the largest habitat area, the highest population density, and largest size, followed by Gongshan muntjac and Fea’s muntjac. This study provides basic data for improving the background knowledge of the animal diversity in the Eastern Himalayan biodiversity hotspot, as well as detailed information and references required by wildlife workers for species identification.

## 1. Introduction

The genus *Muntiacus* belongs to the family Cervidae of the order Artiodactyla. This genus consists of small and medium-sized ungulate mammals that are typical throughout their distribution in East Asia, Southeast Asia, the Indo-China Peninsula, the Malay Peninsula, the Indonesian Sunda Islands, the Indian Subcontinent, and the Himalayas. *Muntiacus* live in tropical and subtropical regions, have large population sizes, and have a large number of species and subspecies; however, the various morphological characteristics of the subclassifications of this genus are not very obvious, making it difficult to identify them in the field [1,2]. According to the taxonomy described in the third edition of “Mammal Species of the World: A Taxonomic and Geographic Reference”, the International Union for Conservation of Nature (IUCN) has assessed 13 species of the genus *Muntiacus* worldwide [3,4,5,6,7,8]. Nevertheless, regardless of whether the existing assessments were based on morphological classification or molecular biological classification, the genus *Muntiacus* has not yet been clearly classified at the species level [5,6,7,8,9,10,11]. Based on this, we believe that it is necessary to resolve the biological and ecological characteristics of muntjacs in the Yarlung Zangbo Grand Canyon National Reserve, a typical region of the Eastern Himalayas, to improve the current understanding of muntjac taxonomy.

### 1.1. Muntiacus vaginalis

The species *Muntiacus vaginalis* is commonly known as the red muntjac or barking muntjac. Before 2003, this species was considered a subspecies of the southern red muntjac (*Muntiacus muntjac* Zimmermann, 1780), but it is now considered an independent species, the northern red muntjac. *Muntiacus vaginalis* is distributed across Southeast Asia, the Indian subcontinent, the Himalayas, and southern China [12]. In contrast, the southern red muntjac (*M. muntjac*) is only distributed across the Sunda Islands in Indonesia. The demarcation line between southern and northern red muntjacs is located in the Khokhok Kra Strait, Thailand. Northern red muntjacs live north of the strait, and southern red muntjacs live south of the strait [1,3,4,12,13].

The internal classification of the northern red muntjac requires further refinement, as some researchers believe that there may be multiple cryptic subspecies. The lack of large-scale field surveys, the limited number of existing specimens, and the limited coverage of specimens are the main reasons for the unclear taxonomic status of the red muntjac. Therefore, further studies combining macrobiology and molecular taxonomy are urgently needed to clarify the taxonomy of the various groups and subspecies of the red muntjac [1,4,13]. However, many descriptive tropical and subtropical subspecies classifications based on morphological characteristics, or the existence of polymorphisms have not been accepted in the mainstream view [4,13]. This suggests that the macroscopic understanding of an animal taxon and biomolecular support for its classification are the two current prerequisites for accepting animal taxonomy.

In the 1970s, scholars measured the number of chromosomes of the red muntjac and found that it has 2n = 6 chromosomes for females and 2n = 7 chromosomes for males [14,15], indicating that the red muntjac is the ungulate with the fewest chromosomes. Due to the tandem fusion of chromosomes in muntjacs, the number of chromosomes in different muntjac species is abnormal, making the genus *Muntiacus* an ideal molecular model for studying chromosome evolution. However, due to the lack of field observation data, few scholars have carried out comprehensive and detailed research on the ecology and biology of muntjacs. This lack of information may also be due to limited awareness of different muntjac species and the inability to identify subspecies. Therefore, research progress on *M. vaginalis* has been slow.

### 1.2. Muntiacus gongshanensis

Based on specific species morphology and mtDNA samples of muntjac specimens collected or purchased in the northern part of Gaoligong Mountain and Fugong County, Yunnan, China, Ma et al. (1990) described a new species of muntjac called the Gongshan muntjac (*M. gongshanensis*) [16]. However, controversy exists about the taxonomic status of this species, though the general understanding and recognition of this species tend to be high. Initially, the Gongshan muntjac was considered to be equivalent to or a western subspecies of the black muntjac (*M. crinifrons*), which is endemic to eastern China [16,17,18,19,20,21]. Subsequently, some researchers studied the black muntjac in the border area between southwestern China and northern Myanmar, as well as in the eastern Himalayas of Tibet [22,23]. Finally, it was shown that there are significant differences between the Gongshan muntjac and the black muntjac in terms of external morphology and chromosome structure [2,5,16,24], indicating that the Gongshan muntjac should be an effective taxon (i.e., an independent species) in the genus *Muntiacus*. In terms of chromosome number, female Gongshan muntjacs have 2n = 8 chromosomes, and males have 2n = 9 chromosomes [14]. In 2019, Huang et al. used the camera-trapping method in Tengchong, Yunnan, China, to visually compare photos of the Gongshan muntjac and the red muntjac for the first time in the survey area [25]. Based on the morphological characteristics and basic reported information about these two species, their differences are clearly apparent [25].

### 1.3. Muntiacus feae

Worldwide, there have been fewer than ten articles published on Fea’s muntjac (*Muntiacus feae*), indicating that this muntjac species is extremely rare. The first report of this species was in 1887, when Leonardo Fae collected one individual in the Tenassin Mountains of Myanmar. In 1889, Thomas and Doria set the type specimen and made a simple report, and in 1892, Thomas created a detailed description of this specimen [26]. It was not until 1977 that Grubb published a detailed report on the specimen’s skull, which has been kept in the Natural Museum of Genoa, Italy [27]. In 1914, Gairdner obtained a sample of Fea’s muntjac chamois from the Philippines in the Denassa area of Mon State, Myanmar. In 1957, Sokolov obtained an incomplete skin near Jinping County (Jinping Miao, Yao and Dai Autonomous County) and Pingbian County (now Pingbian Miao Autonomous County) in Honghe Hani and Yi Autonomous Prefecture, Yunnan Province, southern Yunnan, China. However, subsequent investigations have failed to reconfirm the presence of Fea’s muntjac [6,7,28].

Between 1982 and 1983, six specimens of *M. feae* were obtained for the first time in China in the Dongjiu River Valley in Linzhi County (now Bayi District) and the Yigong River Valley in Bomi County, Tibet [7,26]. Later, three specimens were collected from Gaoligong Mountain and Biluo Snow Mountain in Yunnan [16]. Huang et al. (2006) discovered the fusion relationship of chromosomal tandem repeats in the karyotype evolution using the tissues of the Fea’s muntjac and Gongshan muntjac from the Kunming Cell Bank of the Kunming Institute of Zoology, Chinese Academy of Sciences [10]. The results on the karyotype phenotype were consistent with the results of Soma et al. (1987), namely that female Fea’s muntjacs have 2n = 13 chromosomes and males have 2n = 14 chromosomes [8,29]. Based on related articles and animal monographs, there are no existing photos of Fea’s muntjac, and there seem to have been very few reports on this species in the past 40 years. The distribution, population size, and existing status of this species in the wild are unclear.

Based on the existing information on muntjac species, this study used a decade of infrared camera monitoring in the Yarlung Zangbo Grand Canyon National Nature Reserve in Tibet to determine: (1) how many different muntjac species exist in this area; (2) whether the common and different characteristics between different muntjac individuals can be extracted based on the existing muntjac species monitoring data, with the goal of better serving related grassroots forestry work in this area; (3) what the relationship is between the habitat statuses of different muntjac species at the ground scale; and (4) what the population sizes of different muntjac species are in the study area. The results of this study will provide basic data for researchers and conservation management departments, as well as provide corresponding suggestions and countermeasures for protected area management.

## 2. Materials and Methods

### 2.1. Study Area

Yarlung Zangbo Grand Canyon National Nature Reserve (hereinafter referred to as Grand Canyon Reserve; 29°5′48″–30°20′7″ N, 94°45′28″–96°5′39″ E) is located in Linzhi City, southeast Tibet. The administrative reserve area covers an area of 9168 km^2^, extending over 14 townships in the Bayi District, Milin County, Bomi County, and Medog County (Figure 1).

Due to its size, unique topography, climate, and vegetation, the Grand Canyon Reserve features rich biodiversity. The natural geography and biodiversity of the Grand Canyon Reserve are extremely typical and representative. In particular, the southern Himalayas has the most complete vertical vegetation spectrum of mountain ecosystems in China, with five major ecosystem types, from low to high as low-mountain semi-evergreen monsoon forest, middle-mountain evergreen semi-evergreen broad-leaved forest, subalpine evergreen coniferous forest, alpine shrub meadow, and alpine ice margin vegetation, which becomes an outstanding representative of humid mountains in the southern Himalayas [30]. Fagaceae, Lauraceae, Camelliaceae, Magnoliaceae and Araliaceae are dominant in the survey area. There are more than 3971 species of higher angiosperms and 654 species of higher vertebrates. Under the strong influence of the warm and humid airflow of the South Asian monsoon, the climate in the study area is warm and humid, with a maximum annual precipitation of 4600 mm and an average relative humidity of more than 80%, forming an ecosystem with forests as the dominant population. The forest coverage is high, and it is one of the regions with the highest forest reserves in the world. This area has the largest continuous area of complete forest ecosystem in China, and these remaining and rare complete forest ecosystems provide high-quality habitats for rare wild animals [30].

According to existing survey data, there are 15 wild ungulate species in the reserve, including Mishmi takin (*Budorcas taxicolor*), Bhutanese takin (*B. whitei*), red muntjac (*M. vaginalis*), Gongshan muntjac (*M. gongshanensis*), Fea’s muntjac (*M. feae*), red goral (*Naemorhedus baileyi*), Himalayan goral (*N. goral*), Chinese serow (*Capricornis milneedwardsii*), Himalayan serow (*C. thar*), alpine musk deer (*Moschus chrysogaster*), Himalayan musk deer (*M. leucogaster*), black musk deer (*M. fuscus*), forest musk deer (*M. berezovskii*), blue sheep (*Pseudois nayaur*), and wild boar (*Sus scrofa*) [30].

### 2.2. Camera Trap Survey Method

Eleven representative areas of different elevations and vegetation types in the reserve were selected as survey areas. Camera traps were used to detect and record the wild muntjacs in each area. The survey was conducted in six time periods: January–July 2013, October 2013–May 2014, October 2016–April 2017, October 2017–July 2018, August 2018–November 2019, and July 2020–December 2021. A total of 201 camera trap survey sites were deployed, covering an elevation span of 582–3479 m, with a cumulative survey workload of 56,151 camera days (Table 1). All camera trap sites are more than 500 m apart.

Muntjacs were recorded in all 11 survey areas. Of the recorded sites, the lowest altitude was 582 m, and the highest altitude was 3300 m. A total of 1210 independent photos of muntjacs were taken (Table 1). Other ungulates that were identifiable in the sympatric distribution of muntjacs included Bhutanese takin and Mishmi takin.

Infrared cameras (Acorn Ltl 6210; Ltl Acorn, Des Moines, IA, USA) were set up to shoot three consecutive photographs and one 15-s video after each triggering event. The sensitivity of the cameras was set to medium, and the photo imprint feature was turned on. Each camera was equipped with an SD memory card of 32 GB and 12 rechargeable batteries with a capacity of 2800 mAh. The cameras were not physically concealed during camera installation, and no food bait or attractant was placed nearby. After the installation, the field data card was filled out and numbered. For each camera position and time, consecutive photographs of the muntjacs were taken to compare body size, black striping, coat color, and other physical characteristics as well as physical behavior to determine which photographs were of the same individual. Photographs taken at different times were defined as independent photographs (IPs). When a species was photographed with an infrared camera at a single site, the effective detection of the species was recorded, and the number of independent effective photos was recorded. Based on the first photo of the species, photos of the same species (whether the same individual or not) taken at the same site within 30 min were counted as a single detection of the species. Therefore, the number of independent effective photos was unrelated to the number of individuals of the same species taken in a single photo or single detection [30].

The obtained photos were sorted and screened, and the photos of muntjacs were classified according to different monitoring areas and elevations. The classification time was marked for statistical analysis. For the infrared camera photos taken at the same or similar position and time, the body shape, profile, markings, coat color, other physical characteristics, and physical behaviors of the muntjacs were compared to judge whether the animals belonged to the same image capture period.

### 2.3. Camera Trap Data Processing

#### 2.3.1. Principles of Camera Trap Data Identification

Based on the long-term field survey and the camera trap deployment, we determined the preconditions for camera trap photo identification. These preconditions were that the same camera should take images in the daytime, and photos with better visualization should be used for the identification of the species. We used these preconditions to reduce human error due to misidentification, which could have easily stemmed from poor-quality photos and videos taken at night. On this note, different cameras were used to verify animal body color.

#### 2.3.2. Categories of Camera Trap Observations for Independent Sites

When processing the recovered data from the camera traps set in the field, we summarized it into two categories. The first category was that, based on the behaviors observed in the photos and videos, the camera was located in the home area of the animals. In this case, most of the photographed animals were participating in behaviors such as feeding, playing, grooming, mud bathing, drinking water, and resting. Few animals captured were in this category. In contrast, the second category was that the camera was located at the “crossroads” of wild animal activities, where there were many types of animals and a large amount of activity. Most of the animals photographed in this category displayed behaviors such as passing, chasing, playing, and observing.

#### 2.3.3. Wildlife Identification Method via Camera Traps

Based on the photos and videos obtained via the camera traps, we observed and compared different morphological characteristics of the animals, such as their body shape, overall fur color, and behavior. Furthermore, the color, spots, and patterns of the head, ears, back, limbs, tail, and other parts were observed in detail. Subsequently, clear photos of these areas on the animals’ bodies were compared repeatedly for confirmation.

### 2.4. Suitable Habitat

The points obtained from our infrared camera survey resulted in 54 observations of red muntjac, 36 observations of Gongshan muntjac, and 45 observations of Fea’s muntjac. The habitat size and main distribution range of the three muntjac species in the study area were simulated using MaxEnt software (https://biodiversityinformatics.amnh.org/open_source/maxent/ accessed on 17 November 2020). In the model, the sub-suitable habitat index was 0.5–0.7, and the highly suitable habitat index was 0.7–1.0. Nineteen bioclimatic factor variables and 11 other variables (altitude, annual precipitation, solar radiation, annual average temperature, annual maximum temperature, annual minimum temperature, water vapor pressure, wind speed, ground cover type influencing factor, human activity influencing factor, and number of water bodies) were considered. The raster file had a 1 km^2^ resolution. We defined the habitat index to be 0.5–1.0 as the actual distribution habitat of the species, and the simulated data were corrected by adding limiting factors to the field surveys.

### 2.5. Calculation of Population Sizes

The different species have relatively similar coloration and consequently, it was difficult to identify the species of individuals in the independent photos obtained by camera-trapping technology. We used the Random Encounter Method (REM) described by Rowcliffe et al. (2008) to estimate species populations using camera-trapping technology [31]. Assuming the species investigated by the infrared camera are comparable to molecules in a closed space, the moving coverage area in a certain time is composed of molecular linearity and moving distance. Therefore, the expected collision rate between molecules in a specific time range refers to the proportion of the moving coverage area of all molecules to the total area [31,32]. In other words, the number of collisions *y* between a relatively stationary molecule (species) and a circular detection area (the camera’s angle of view) in biology (the total number of independent photos of individual species) is determined by the moving speed *v* of the molecule (species), the moving time *t* (camera working days), the distance *r* the molecules move (the radius of the camera monitoring area), and the molecular number density *D* (species density), as shown in Equation (1).
(1)y=2rtvD

Since the monitoring area of the camera is not a circle, but rather a fan-shaped area composed of a radius *r* and an angle *θ*, the width covered by the captured molecules (species) is not simply 2*r* but includes an angle. The sector area was calculated as (π − *θ*)/2 ≤ γ ≤ π/2, as derived from Rowcliffe et al. (2008) [31].
(2)$$y=\frac{2+\theta }{\pi }rtv\theta$$

We referred to Rowcliffe et al. (2008) for our study of muntjac population density [31]. The statistical *t* value is based on the effective number of hours during the six-week daytime (12 h) camera working period. The following formula (Equation (3)) was thus obtained:(3)$$MCV=\frac{y}{t}$$
where *MCV* is the annual shooting value, *y* is the number of independent photos of an individual muntjac in that year, and *t* is the number of days in which the camera was deployed in that year (Equation (4)).
(4)$$D=MCV\frac{\pi }{vr\left(2+\theta \right)}$$

The monitoring angle *θ* of the infrared camera used was 0.872 rad (50°), and the camera monitoring radius *r* was 0.01 km. The muntjacs in the field were found to have a daily activity distance of 1–3 km. The daily movement speed of the muntjacs was set according to three parameters: *v*_1_ (1 km/d), *v*_2_ (2 km/d), and *v*_3_ (3 km/d).

## 3. Results

### 3.1. Morphological Identification and Common Features of the Three Muntjac Species

Among the 1210 independent photos of muntjacs obtained in this camera trap survey, 540 were taken during the day, accounting for 44.63% of the total number of independent photos. These photos were used for morphological recognition. A total of three different muntjac species were recorded, namely the red muntjac, Gongshan muntjac, and Fea’s muntjac (Figure 2). The 1210 independent photos of red muntjacs, Gongshan muntjacs, and Fea’s muntjacs were investigated by referring to relevant materials and infrared camera technology. Of the 540 individual photos taken during the day, 328 were of red muntjacs, accounting for 60.74% of the total identifiable photos; 203 photos were of Gongshan muntjacs, accounting for 37.59% of the total identifiable photos; and 9 photos were of Fea’s muntjacs, accounting for 1.67% of the total identifiable photos. Relevant professional animal books, network databases, laboratory specimens, from details (head, ear, limbs, tail) to the whole species were consulted to permit identification.

The differences in the morphological characteristics of the three muntjac species are described as follows:

Red muntjac (*M. vaginalis*): The face is relatively long and narrow, and there is a broad and prominent frontal gland on each side, from the infraorbital gland to the horn bifurcation. The frontal glands are long and eventually cross together to form a blackish “V” shape on the snout. Adults are tall with a body length between 90–120 cm. The whole body is orange; the lower jaw, the inside of the limbs, the abdomen, and the undertail are milky white; and the slender limbs are proportional to the body. The tail is an inverted triangle with a milky white periphery. Males have short single-forked horns that extend straight and backward, with long horn bases, inwardly curved horn tips, and opposite bicuspids. Females have no horns, but the top of the forehead is slightly protruding with special black hairs formed in bundles, corresponding to the same part of the body where the horns are located in males. In terms of both body color and size, the red muntjac is quite different from the Gongshan muntjac and Fea’s muntjac (Figure 2a–d). Using the independent site whole-event observation method, we observed that the limbs of sexually immature individuals are orange, and with increased sexual maturity and age, the limbs gradually turn black, with females having a weaker color than males.

Gongshan muntjac (*M. gongshanensis*): In adults, the body length is 70–100 cm with a body size significantly smaller than that of the red muntjac. Gongshan muntjacs are dark brown with black limbs, buttocks, tail undersides, and tail interiors. The tail edge is prominently milky white. The color between the forehead and the muzzle is slightly black, and the color from the infraorbital gland to the muzzle is black. The photos and videos showed that the top of the head and the base of the horns are orange–yellow, like the red muntjac. Males have horns and females have no horns. Overall, body size, color, and behavior are easily distinguishable from the red muntjac (Figure 2e–h).

Fea’s muntjac (*Muntiacus feae*): The size of this muntjac is equal to that of the red muntjac and significantly larger than that of the Gongshan muntjac. The coat color is intermediate between the two other species. The overall color is light brown or dark orange. The orbital glands under the eyes to the muzzle are black, and the limbs and the back of the tail are black, similar to the Gongshan muntjac. In females, the biggest difference compared to the red muntjac is that the tail is black, whereas the tail is orange in the red muntjac. The biggest difference from the Gongshan muntjac is that this species is larger, and its black coloration is limited to the limbs, back of the tail, and part of the face. Using the independent site whole-event observation method, the three muntjac species are easily distinguished (Figure 2i–l).

### 3.2. Suitable Habitats for the Three Muntjac Species

According to the ROC curve output by the model, AUC values of all models are greater than 0.98, indicating that the model prediction has achieved good accuracy. According to the MaxEnt niche model, the most suitable habitat area for red muntjac was 4292 km^2^, and the suitable habitat distribution area was 9140 km^2^. The most suitable habitat area for Gongshan muntjac was 2502 km^2^, and the suitable habitat distribution area was 9459 km^2^. The most suitable habitat area for Fea’s muntjac was 2552 km^2^, and the suitable habitat distribution area was 5657 km^2^.

From the simulation results, it can be concluded that the most suitable habitat area ratio in the study area was red muntjac: Gongshan muntjac: Fea’s muntjac = 1:0.58:0.59. The suitable habitat distribution area ratio was red muntjac: Gongshan muntjac: Fea’s muntjac = 1:1.03:0.62.

Based on the simulation results, we concluded that Gongshan muntjac and Fea’s muntjac are distributed on the east and west banks of the Yarlung Zangbo River as the boundary; however, the actual camera trap survey presented the illusion that there were fewer Fea’s muntjac than in reality. Most of the main research areas are uninhabited areas that are difficult to reach; thus, the model reveals that there are more areas that are suitable habitats. Among them, the distribution inflection points of red muntjac and Gongshan muntjac are in Damu Township, Medog County (1550 m above sea level), and the distribution range of the Gongshan muntjac is in the east Damu Township–Gedang Township–Jinzhu Zangbo Watershed. The red muntjac is distributed on both sides of the Yarlung Zangbo River along the line of Damu Township–Medog County (1091 m above sea level)–Beibeng Township (843 m above sea level). The red muntjac is distributed along the east and west banks of the Yarlung Zangbo River in Beibeng Township–Baima West Road River–Khanmi (2123 m above sea level). According to the simulation results of the MaxEnt niche model, the habitat index is 0.7–1.0, the niche differentiation of the three muntjac species is obvious, and there is no high overlap of their core habitats. The overlapping ratio of red muntjac: Gongshan muntjac: Fea’s muntjac was 1:0.05:0.001. When the habitat index was 0.5–1.0, the overlapping ratio of red muntjac: Gongshan muntjac: Fea’s muntjac was 1:0.13:0.07. It can be seen that the three muntjac species in the study area are significantly differentiated in terms of habitat (Figure 3).

### 3.3. Populations of the Three Muntjac Species

Based on the moving speed *v*_1_ (1 km/d), *v*_2_ (2 km/d), and *v*_3_ (3 km/d) parameters of the three species, the densities of these species are D_1_, D_2_, and D_3_ (Table 2). There was a highly significant difference in the unit density of red muntjac at different moving speeds (F = 1.579, df = 2, *p* < 0.01), there was no significant difference in the unit density of Gongshan muntjac at different moving speeds (F = 1.1105, df = 2, *p* > 0.05), and there was a highly significant difference in the unit density of Fea’s muntjacs at different moving speeds (F = 0.5129, df = 2, *p* < 0.01).

There were no significant differences in population density using the least significant difference method (LSD), so the actual density is subject to the average density. Considering that the actual survey area is physically inaccessible to researchers, data is lacking for some research areas. The total protected area of 9168 km^2^ was used to calculate the area of the population. The population density of red muntjac in the study area was 0.8495 ± 0.4217 individuals per square kilometer, and the population size was 7788 ± 3866. The population density of Gongshan muntjac in the study area was 0.7279 ± 0.2314 individuals per square kilometer, and the population size was 6673 ± 2121. The population density of Fea’s muntjac in the study area was 0.3428 ± 0.1027 individuals per square kilometer, and the population size was 3142 ± 942.

## 4. Discussion

The habitats of species are complex, overlapping, and changing, in part due to the existence of the habitat transition zone between the Palaearctic and the Orient [33]. This complexity is evidenced by the long-time lack of a systematic and comprehensive survey of animal background resources in the study area. Most ungulate species in the study area coexist with their sister species and cryptic species. In addition, due to the continuous changes in mammalian classification systems, the awareness of the species in our study region has undoubtedly increased. Based on 10 years of monitoring data combined with the results of the second terrestrial wildlife resource survey project in the Tibet Autonomous Region, we used visual photos to investigate the red muntjac, Gongshan muntjac, and Fea’s muntjac in China’s Tibet Grand Canyon National Nature Reserve. The morphological characteristics, habitat distribution range, and population sizes of the three muntjac species in the region were estimated, answering many questions that have long puzzled researchers in the study area.

With the continuous development of science and technology, conservation biology has incorporated newer technical means, such as infrared camera trap technology, “3S” technology, satellite collars, wildlife DNA metagenomic sequencing, and other technologies. The introduction of these technologies has greatly promoted the rapid development of conservation biology and zoology, and it has also led to the gradual development of conservation biology towards quantification and refinement [9,34,35,36,37]. In 2020, Hu et al. used genome sequencing technology to resolve the long-standing subspecies controversy of the red panda [37]. They clarified the evolutionary history of the two known subspecies through genome sequencing and zoogeography, and then classified them into separate species, the Himalayan red panda (*Ailurus fulgens*) and the Chinese red panda (*A. styani*) [38]. In 2015, Li et al. published a new species, the white-cheeked macaque (*Macaca leucogenys*), by using visual camera-trap photos as positive and subtype specimens, comparing physical appearance features, and using animal sound analysis technology [39]. We believe that species identification should start from the aspects of appearance, habitat, population size, distribution range, habits, and behaviors. First, understanding the basic information of the species itself and its living environment is a necessary prerequisite for understanding the species [30,40,41,42]. This study uses the long-term accumulation of visual data to resolve the ambiguity of researchers’ understanding of muntjac species. When mistakes are made in species identification, the obtained research results are unscientific and may cause ambiguity and disputes at the academic level. In this respect, the correct knowledge of species is of paramount importance.

The species in the survey area and their distribution ranges and population sizes have persistently been the primary concerns of researchers. Compared with traditional surveys, infrared camera trap technology is ineffective for understanding the number of species in a given area [40,41,42]. However, it can be used to not only provide researchers with visual photos and videos, but also to continuously explore and study the activity boundaries of animals and population ecology issues such as population size. With the integration of mathematical models, the long-term accumulation of species visualizations will provide key breakthroughs in addressing population ecology issues. For example, in the past three decades, the MaxEnt niche model has been widely used in conservation biology, invasion biology, and applied ecology. Consequently, the actual and potential distribution areas of plant species have been widely recognized by scholars [40,41,42,43,44]. The “potential suitable distribution area” of the Maxent model usually refers to the area and suitability level of potential suitable habitat without considering the impact of species migration capacity, interaction between species, epidemic disease, poaching and other factors on species, and only considers the internal relationship between species and environment. The purpose of using the model is not to prove that the occupied habitat is a potential distribution area, but to explain what other potential habitats are, their area size, quality, and what factors affect them. On the one hand, the accuracy of the model can be proved by mathematical methods, such as AUC size. On the other hand, it can be compared with existing literature or verified on the spot, but it should be noted that the relationship between species and between species and the environment is very complex. Although the model is a good method, it is not enough to completely simulate the natural ecosystem. It is also difficult to comprehensively assess the quality and actual size of the habitat through human field verification, so the simulation results still have great reference value [40,43,44].

In the random encounter model in this study, we adjusted the parameters including speed, range of motion and direction of motion to improve the accuracy of the model. At the same time, the process of being photographed in different motion modes is simulated, and a model is established to quantify the relationship between population density and the number of photos using 10 years of monitoring data. Based on the population distribution, quantity and other relevant species parameters of the Second Terrestrial Wildlife Resources Survey of Tibet Autonomous Region completed in 2020, the density estimation and confidence interval were compared. The survey is calculated by using the traditional sample line and sample point method data. Of course, the random encounter model is controversial in estimating the number of species, but on the basis of controlling relevant variables and combining with field verification, the accuracy of the model’s population density is still worth our attempt [31,32]. The integrated application of new technologies plays a key role in understanding species in complex environments and strongly promotes the rapid development of conservation biology. Under the framework of the main branch of taxonomy, that is, the mainstream classification of muntjac animals, this paper maintains consistent research results. Due to the lack of biomolecular support, the following points require future study: (1) whether there are cryptic species of red muntjac; (2) whether there are other new or effective muntjac species in this area; (3) further verification and exploration of the biological and ecological problems related to Fea’s muntjac; (4) the relationship between Gongshan muntjac and Yunnan Gaoligongshan populations in the study area; and (5) the differentiation mechanism of the three muntjac species in the study area and the internal connections of their molecular biology.

Since China’s Tibet Grand Canyon National Nature Reserve is located in the Eastern Himalayan hotspot, the region is extremely rich in animal and plant resources. It is believed that with the continuous development of scientific and technological means and the accumulation of data from scientific and technical workers, more information on the muntjac species in this area can be obtained in the near future.

## 5. Conclusions

From 2013 to 2022, camera-trapping was employed to monitor the wild ungulates in the Yarlung Zangbo Grand Canyon National Nature Reserve in Tibet. Through the identification and analysis of the ten years of collected data, the first field photo of Fea’s muntjac since its naming in 1887 was obtained. At the same time, the external morphology of three species of muntjac, namely the red muntjac, Gongshan muntjac, and Fea’s muntjac, were distinguished based on the photos and texts, and these species are visually presented to the readers. On the basis of obtaining reliable activity points, the MaxEnt model was used to simulate the habitats of the three muntjac species. The red muntjac had the largest habitat area in the study site, followed by Gongshan muntjac and Fea’s muntjac. Furthermore, the Random Encounter Model (REM) was used to estimate the density and population size of the three species of muntjac in the reserve. The population density of red muntjac in the study area was 0.8495 ± 0.4217 individuals/km^2^, and the population number was 7788 ± 3866 individuals. The population density of Gongshan muntjac in the study area was 0.7279 ± 0.2314 individuals/km^2^, and the population number was 6673 ± 2121 individuals. The population density of Fea’s muntjac in the study area was 0.3428 ± 0.1027 individuals/km^2^, and the population number was 3142 ± 942 individuals. The research results of the above model may indicate that the larger muntjac species in the study area have more habitats and higher population numbers, which indicates that there is a certain correlation between the degree of competition among ungulates and their size in the area, that is, the larger muntjac species have greater pressure to survive in the area than the smaller muntjac species, and the various disturbances they face generate the advantages of population and habitat. Regardless of the population size or the habitat occupied, large muntjac species have certain advantages and more options. To a certain extent, the basic animal ecology of these three muntjac species in the study area was revealed.

## Figures and Tables

**Figure 1 animals-12-02909-f001:**
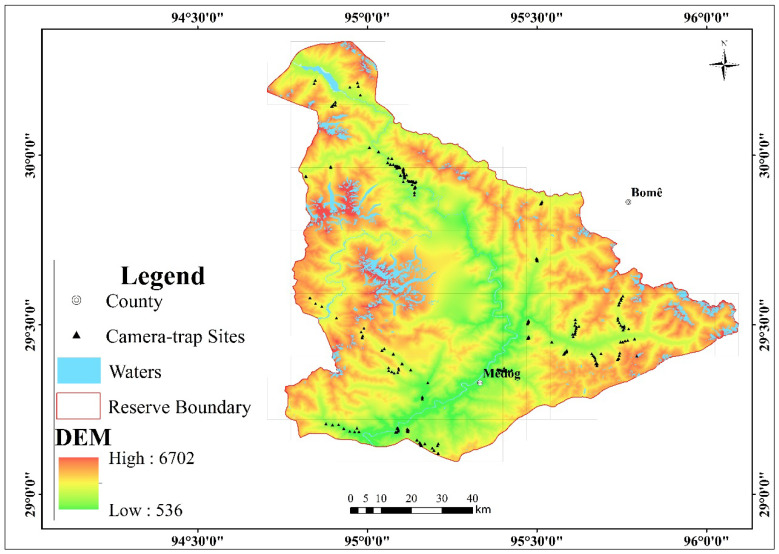
Study area and camera trap sites in the Yarlung Zangbo Grand Canyon National Nature Reserve, Tibet, from 2013 to 2021.

**Figure 2 animals-12-02909-f002:**
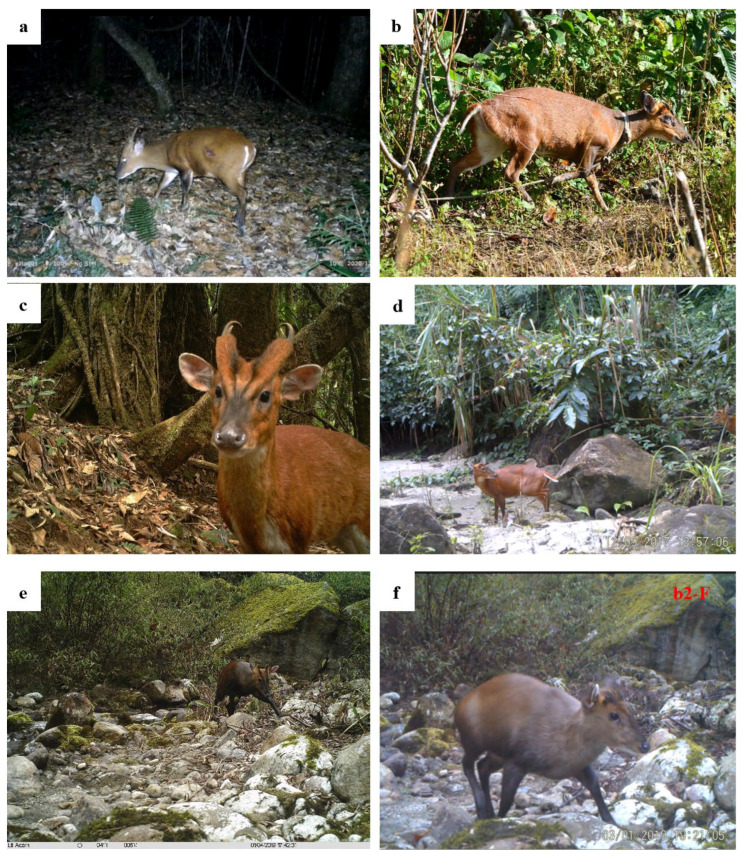
The three muntjac species present in the Tibetan Yarlung Zangbo Grand Canyon National Nature Reserve. (**a**–**d**) Red muntjac (*Muntiacus vaginalis*); (**e**–**h**) Gongshan muntjac (*M. gongshanensis*); (**i**–**l**) Fea’s muntjac (*M. feae*). Note: (**a**,**c**,**e**,**g**,**i**) are male, and (**b**,**d**,**f**,**h**,**j**–**l**) are female. The red arrows pointed at the tails indicate the biggest difference between *M. feae* and *M. vaginalis*.

**Figure 3 animals-12-02909-f003:**
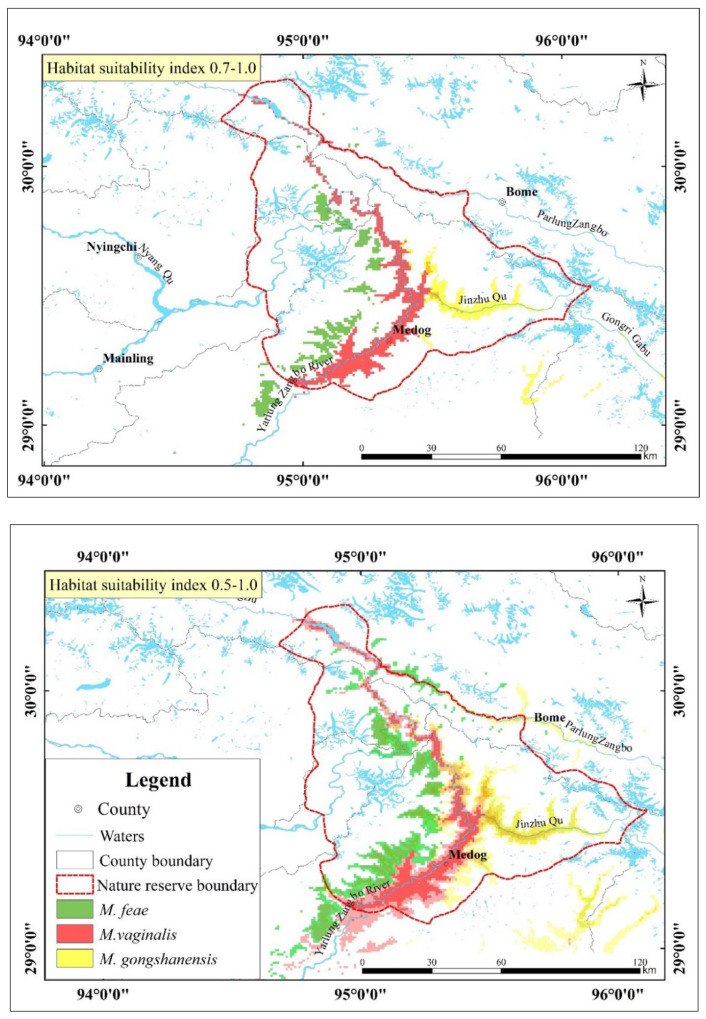
Potential distribution areas of the three muntjac species the Yarlung Zangbo Grand Canyon National Nature Reserve, Tibet.

**Table 1 animals-12-02909-t001:** Camera trap efforts and number of independent photographs of *Muntiacus* in the 11 survey areas in the Yarlung Zangbo Grand Canyon National Nature Reserve, Tibet, from 2013 to 2021.

Survey Area	Number of Camera Stations	Elevation Range	Number of Camera Days	Number of Photographs	Number of Independent Photographs of *Muntiacus*
Bi Xi Ri Area	11	2235–3479 m	2794	12,294	15
South bank of the Yarlung Zangbo River Area	10	582–668 m	1880	16,751	13
Uma Mountain Area	6	1751–3145 m	1374	6942	34
Raj Mountain Area	8	1631–2086 m	1968	8467	28
Dan Ge Zhuo Area	3	954–1434 m	630	684	13
Ge Dang Ditch Area	36	2230–4470 m	8691	10,713	429
Mei Yu Lun Ba Area	2	1751–2315 m	294	5172	19
Xi Gong River Area	6	1124–1590 m	1080	6532	22
De Yang Ditch Area	8	815–1294 m	1360	6359	23
North of the Grand Canyon	31	3650–4700 m	5580	4578	0
De Er Gong Area	80	1750–2890 m	30,500	11,980	614
Total	201	815–4700 m	56,151	90,472	1210

**Table 2 animals-12-02909-t002:** *Muntiacus* population densities estimated by the infrared triggered camera technique in the Yarlung Zangbo Grand Canyon National Nature Reserve, Tibet, from 2013 to 2021.

*Muntiacus*	IP	Camera Days	MCV	D_1_	D_2_	D_3_	Mean D ± SE
*M. vaginalis*	605	37,451	0.0162	1.7661	0.5956	0.1869	0.8495 ± 0.4217
*M. gongshanensis*	416	38,180	0.0109	1.1912	0.5956	0.3970	0.7279 ± 0.2314
*M. feae*	189	36,842	0.0051	0.5609	0.2804	0.1870	0.3428 ± 0.1027

IP: number of independent photos; MCV: annual shooting value; D_1_, D_2_, and D_3_: densities based on moving speed *v*_1_ (1 km/d), *v*_2_ (2 km/d), and *v*_3_ (3 km/d), respectively.

## Data Availability

The data presented in this study are available on request from the corresponding author.

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
