# Peer review of "Morphological, Phaneroptic, Habitat and Population Description of Three Muntjac Species in a Tibetan Nature Reserve"

_animals, 2022, doi:10.3390/ani12212909_

Round 1

Reviewer 1 Report

This article used valuable long-term data collected by camera trappers, however, many details of data processing and analyses need to be provided or clarified. My comments and suggestions:

1.   Sentences in Line 194-196, Line 197-198, Line 199-202, Line 210-211, Line 226-227, Line 243-245, Line 333, Line 379-382 were not clear or difficult to understand, re-write these sentences are necessary.

2.   For species identification, only 1 person involved in the process or 2 or more people involved? Did you double check the result of the identification? If yes, please explain how did you verify its accuracy in the Materials and Methods section, so the quality of muntjac species identification based on photo image can be trust.

3.   The whole section of 2.3.2 can be removed.

4.   Please define the “observations” in Line 229-230, and explain their differences with the “number of photo taken” in Line 334-337, and with the “IP (the number of independent photos)” in Table 2. They were all different.

5.   Please double check the accuracy of the equation (2) (Line 262).

6.   Please provide references for the daily moving distances for muntjac species mentioned in Line 272-273.

7.   What is “independent site whole-event observation method (Line 304)”? Any references?

8.   Please add, in the Result section, the details of the Maxent models produced for the three muntjac species (e.g., variables included in the best model, quality of the best model, the threshold for the best model…etc.).

9.   In Table 2, why “Camera days” were different for different muntjac species? And, why they were different from the 56,151 camera days showed in Table 1. Also, when calculated the MCV values, you should use 1/2 camera days, right? Because only daytime pictures were considered in this study, therefore if using the whole day to calculate MCV will under-estimate the population size.

10.     The “Discussion” section discusses only a few information presented in the Result section.

Reviewer 2 Report

To my concern the ms. is sound in all of this sections. Perhaps, the Discussion section could be improved by the discussing the limitations of the study (e.g., accuracy of density estimations, the bias toward physical variables when using Maxent. Despite this, I rate high the 10 years of study in a very harsh environment and the use of over 200 camera traps.

Reviewer 3 Report

It is a work of interest for this animal species since it is a wild genetic resource, which, although there are studies, the same authors mention that they are scarce in the studied species.

 These are the observations

 Title

I suggest changing the title to: “Morphological, phaneroptic, habitat and population description of three muntjac species in a Tibetan nature reserve”

 ABSTRACT

The abstract must be structured so that the reader understands it. I consider that the word taxonomy should be eliminated throughout the text, what they did was a brief morphological description.

The objective of the research is not described in the abstract

They must write the number of animals per species, axis Muntiacus vaginalis (n=?). the conclusion is missing

 Keywords

Write keywords that allow you to search for the topic in indexing systems and in databases. For example: conservation, wild species...

 Introduction

Line 52 to 53. Write the data of the population (if you have it) You must place a reference

 Line 141 to 151. The authors mention that monitoring was carried out for a decade to determine 1) how many species, 2) common characteristics, 3) habitat relationship between species and 4) population size. However, it is not clear what the general objective of the research is. It is recommended to formulate an objective that illustrates the reader about the purpose of the investigation.

 Materials and methods

Line 162 to 163: it is necessary to place a bibliographic source

 Line 163 to 168. I would recommend writing the estimated number of animals per species. However, since this is a study on the Muntjac genus, you should only list the species of this genus, and list the estimated population per species, if you have any data.

It is recommended to make a more detailed description of the site, including relative humidity, temperature, plant species,

 line 173 to 178. What was the distance between camera and camera?

 line 179. Table 1 should be placed here

 Results

line 280. The morphological recognition must be very clear, since morphology studies the shape of the animal. What were the criteria for morphological recognition?

Note that in lines 243 to 244 it says that the population size could not be calculated because the species have similar coloration; I wonder: ¿how did they recognize those three different species?

 It should be explained in materials and methods

 Line 29. ...morphological, phaneroptic.....

 Line 290 to 324. In materials and methods they should write what were the characteristics, phaneroptic, morphological that they were going to describe; length of the face, length of the body, height of the animal, color of the hair, etc.

 line 195: how did you get these lengths? Did you estimate them in the photographs? please explain

 Discussion

Line 395. ..."transition zone between the Palaeartctic and the Oriente"; notice how important it is to describe the study site. This description must be detailed in the study site

 line 401. What second project? I don't understand

 Line 407 to 419. I consider that this paragraph is not for discussion. This paragraph should be in the introduction

 Conclusions

Line 456 to 472. The authors present part of the results as a conclusion. Authors are recommended to write a concrete conclusion on the topic discussed describing the most important findings. It is recommended to correct

Round 2

Reviewer 3 Report

The authors did not address the referee's comments. The manuscript has not been enhanced for publication

I am attaching the comments again

Round 3

Reviewer 3 Report

I recommend removing the word taxonomists

Author Response

As per the Reviewer's comment, we have revised the relevant word.